# Postbiotics Derived from *L. paracasei* ET-22 Inhibit the Formation of *S. mutans* Biofilms and Bioactive Substances: An Analysis

**DOI:** 10.3390/molecules28031236

**Published:** 2023-01-27

**Authors:** Zhi Zhao, Jianmin Wu, Zhe Sun, Jinbo Fan, Fudong Liu, Wen Zhao, Wei-Hsien Liu, Ming Zhang, Wei-Lian Hung

**Affiliations:** 1School of Food and Health, Beijing Technology and Business University, Beijing 100024, China; 2College of Food Science and Engineering, Bohai University, Jinzhou 121013, China; 3China Key Laboratory of Precision Nutrition and Food Quality, Department of Nutrition and Health, China Agricultural University, Beijing 100083, China; 4Inner Mongolia Dairy Technology Research Institute Co., Ltd., Hohhot 010110, China; 5Inner Mongolia Yili Industrial Group Co., Ltd., Hohhot 010110, China

**Keywords:** dental caries, *L. paracasei* ET-22, postbiotics, biofilm, *S. mutans*

## Abstract

Globally, dental caries is one of the most common non-communicable diseases for patients of all ages; *Streptococcus mutans* (*S. mutans*) is its principal pathogen. *Lactobacillus paracasei* (*L. paracasei*) shows excellent anti-pathogens and immune-regulation functions in the host. The aim of this study is to evaluate the effects of *L. paracasei* ET-22 on the formation of *S. mutans* biofilms. The living bacteria, heat-killed bacteria, and secretions of *L. paracasei* ET-22 were prepared using the same number of bacteria. In vitro, they were added into artificial-saliva medium, and used to coculture with the *S. mutans*. Results showed that the living bacteria and secretions of *L. paracasei* ET-22 inhibited biofilm-growth, the synthesis of water-soluble polysaccharide and water-insoluble polysaccharide, and virulence-gene-expression levels related to the formation of *S. mutans* biofilms. Surprisingly, the heat-killed *L. paracasei* ET-22, which is a postbiotic, also showed a similar regulation function. Non-targeted metabonomics technology was used to identify multiple potential active-substances in the postbiotics of *L. paracasei* ET-22 that inhibit the formation of *S. mutans* biofilms, including phenyllactic acid, zidovudine monophosphate, and citrulline. In conclusion, live bacteria and its postbiotics of *L. paracasei* ET-22 all have inhibitory effects on the formation of *S. mutans* biofilm. The postbiotics of *L. paracasei* ET-22 may be a promising biological anticariogenic-agent.

## 1. Introduction

For patients of all ages, dental caries is one of the most common non-communicable diseases worldwide [1]. It is a multifactorial chronic disease caused by chronic biofilms (plaques) containing cariogenic microorganisms. These microorganisms and their virulence products collectively cause the demineralization and progressive destruction of enamel [1,2].

As a common pathogen of oral microbiota, *Streptococcus mutans* (*S. mutans*) has long been recognized as the primary microorganism that causes dental caries [3]. The biofilm is a highly dynamic and structured community of microbial cells integrated into a self-produced extracellular polymer matrix [4,5]. A certain number of *S. mutans* and other bacteria can coordinate the formation of three-dimensional cariogenic biofilms by producing extracellular polysaccharides (EPS) in the extracellular matrix, creating a microenvironment rich in pathogens and fermentable dietary carbohydrates [2,6]. In this microenvironment, metabolites such as acids produced by *S. mutans* accumulate in the biofilms and at the interface between the biofilms and the tooth surface, causing enamel demineralization and then the dental caries [7].

Although most dental plaques can be removed, residual dental plaques are inevitable [1]. At present, the main means of preventing dental caries are the administration of fluoride and avidin and mechanical removal [8,9]. However, each of these therapies has its limitations. Mechanical removal by brushing and flossing might be considered the first line of prevention, as it is mainly used in the daily maintenance of oral hygiene [10]. However, strongly adherent biofilms may not be cleared out completely. Furthermore, some dead spaces cannot be touched. Even though fluoride can affect the glycolytic activity of *Streptococcus*, its ability to protect against dental caries is insufficient and it has little impact on microorganisms’ ability to produce acid [11]. As for avidin, although it effectively prevents dental caries in in vitro and in vivo experiments, its excessive use changes the microbiota of the digestive tract and then increases the drug resistance of pathogens [12]. Therefore, more convenient, efficient, and safe methods are needed to inhibit the formation of dental plaques.

Probiotics maintain oral health by promoting a microbial balance [9]. Probiotics can prevent caries by producing metabolites, such as biosurfactants, bacteriocin, and EPS, inhibiting adhesion and colonization, and downregulating the expressions of virulence genes related to the biofilm formation of cariogenic pathogens [13]. Lactobacillus is a symbiotic microorganism found in the human oral microbiota, and has a strong impact on *Streptococcus* in the oral cavity [14]. Meanwhile, long-term studies also found that heat-killed probiotics, known as postbiotics, also show beneficial effects [15]. This may be another pathway for using supplemented probiotics to perform regulated functions. Moreover, the effect of postbiotics may be more stable and superior in some application scenarios [16]. In Lactobacillus, *L. paracasei, L. plantarum, L. alivarius*, and *L. rhamnosus* are the most used species for digestive-system health [14]. However, the effect mechanisms of *Lactobacillus* on oral *S. mutans* have not been fully clarified. LC-MS/MS is an important technique for analyzing the functional substances of postbiotics. In addition, within the same species, bacterial bioactivity is strain specific. Therefore, the mechanisms of regulation may be disparate and worth interrogating further.

The purpose of this study was to evaluate the effects of *L. paracasei* ET-22 on cariogenic biofilms and investigate the potential mechanisms of *L. paracasei* ET-22 in the intervention of cariogenic biofilms. In vitro experiments were used to conduct preliminary investigations of the effects of live bacteria, heat-killed bacteria, and secretions of *L. paracasei* ET-22 on the formation of *S. mutans* biofilms and functional mechanisms.

## 2. Results

### 2.1. Effects of L. paracasei ET-22 on the Biofilm Formation and EPS Production of S. mutans

To assess the effects of *L. paracasei* ET-22 on *S. mutans* biofilms, the live bacteria, heat-killed bacteria, and secretions of *L. paracasei* ET-22 were used in a co-culture with *S. mutans*. An equal volume of bacterial solutions with a concentration of 5.5 × 10^8^ CFU/mL was used to prepare the live bacteria (ET-22-V), heat-killed bacteria (ET-22-HK), and secretions (ET-22-S) of *L. paracasei* ET-22. The single pathogen *S. mutans* (Model) was cultured as a control. After a 24 h co-culture with *S. mutans*, the ET-22-V and ET-22-S groups all showed decreased crystal-violet-staining reading values (*p* < 0.01, Figure 1A). Surprisingly, we found that the ET-22-HK group also showed a reduced crystal-violet-staining reading value (*p* < 0.01, Figure 1A).

Based on the inhibitory effects of the live bacteria, heat-killed bacteria, and secretions of *L. paracasei* ET-22 on *S. mutans* biofilms, we further analyzed the EPS contents of *S. mutans* biofilms. Compared with the Model group, the ET-22-V, ET-22-HK, and ET-22-S groups all showed significantly decreased contents of insoluble EPS in *S. mutans* biofilms (*p* < 0.01, Figure 1B). Meanwhile, the ET-22-V, ET-22-HK, and ET-22-S groups all showed significantly reduced contents of soluble EPS in *S. mutans* biofilms compared to the Model group (*p* < 0.01, Figure 1C).

### 2.2. Biofilm Microstructure Is Changed by L. paracasei ET-22

To further verify the bacteria’s inhibitory effects for *S. mutans* biofilms, we viewed its microstructure using scanning electron microscopy (SEM) under the 10 μm or 5 μm scales. According to the representative SEM images, the Model group showed a stable, multilayer, and reticular biofilm, in which the *S. mutans* was tightly wrapped with a cauliflower-like structure (Figure 2(A-1)). A small number of bacteria were exposed, and closely adhered to the “flower stalk” (black arrow) and “corolla” (red arrow) positions (Figure 2(A-2)). The ET-22-V, ET-22-HK and ET-22-S groups all showed obviously changed biofilm microstructures (Figure 2B–D). In the ET-22-V group, the multi-layer and compact cauliflower-like structure disappeared. Instead, a single-layer and loose biofilm-microstructure was exhibited, in which the bacilliform *L. paracasei* ET-22 and spherical *S. mutans* were completely exposed on the biofilms. The bindings of these two strains with the biofilms were loose (Figure 2B). Similarly, the ET-22-HK and ET-22-S groups also presented a lack of cauliflower-like structures (Figure 2C,D). In the ET-22-HK group, the heat-killed *L. paracasei* ET-22 and spherical *S. mutans* were tightly wrapped into a single-layer sheet structure (Figure 2C). The ET-22-S group showed a compact and discontinuous biofilm with a certain thickness, in which the *S. mutans* were tightly wrapped (Figure 2D).

### 2.3. Biofilm Thickness Is Changed by L. paracasei ET-22

We further detected the biofilm thickness using confocal laser scanning microscopy (CLSM). Using the COMSTAT 2 analysis for CLSM views, the biofilm thicknesses are displayed. As shown in Figure 3A, compared with the Model group, the thicknesses of the *S. mutans* biofilms were significantly lower in the ET-22-V, ET-22-HK, and ET-22-S groups (*p* < 0.01, Figure 3A). After being stained with N01 and propidium iodide (PI), the live bacteria (green) and dead bacteria (red) in the biofilms were viewed (Figure 3B). Based on the COMSTAT 2 analysis, living and dead cells were respectively quantified. After the co-culture, the biomasses of live bacteria in the biofilms of the ET-22-V, ET-22-HK, and ET-22-S groups were remarkably lower than those in the Model group (*p* < 0.01, Figure 3C). However, there was no significant difference in the biomasses of dead bacteria among these groups (*p* > 0.05, Figure 3C). The vertical biofilm-coverage was checked. In the Model group, both the living bacteria and the dead bacteria in the biofilms gathered mainly at the 50–115 μm and 15–40 μm positions, above the bottom of the hydroxyapatite discs (Figure 3D,E). By contrast, the living bacteria and dead bacteria in the biofilms of the ET-22-V, ET-22-HK, and ET-22-S groups mainly gathered at the 15–40 μm position (Figure 3D,E).

### 2.4. The Expression of Virulence Genes and the Quorum-Sensing System Are Regulated by L. paracasei ET-22

In order to investigate the mechanisms that inhibit the formation of *S. mutans* biofilms, the gene-expression levels of virulence-factors and quorum-sensing (QS) systems were checked. brpA is involved in the formation of biofilms and the stress tolerance of *S. mutans*. *SpaP* is a non-sucrose-dependent adhesion gene. Its protein SPAP mediates the initial adhesion of *S. mutans* to teeth. *S. mutans* also synthesizes some surface-associated glucan-binding proteins (GBP), including gbpA, gbpB, gbpC, and gbpD, and glucosyltransferases including gtfB, gtfC, and gtfD, contributing to the formation of biofilms. After the cotreatment, the ET-22-V, ET-22-HK, and ET-22-S groups all presented significantly decreased gene-expression-levels of *brpA*, *SpaP*, *gbpB*, *gbpC*, *gbpD*, and *gtfB*, compared to the Model group (*p* < 0.05, Figure 4A).

Lactate dehydrogenase (LDH), rela, RecA, and ffh are all involved in the acid production and acid tolerance of *S. mutans*. Compared with the Model group, the gene expressions of *LDH*, *rela, RecA*, and *ffh* were significantly reduced in the ET-22-V, ET-22-HK, and ET-22-S groups (*p* < 0.01, Figure 4B). As a component of the QS system, ComDE can regulate the intraspecific communications of *S. mutans* in biofilms by stimulating the competitive signaling-peptides. Compared with the Model group, the ET-22-V and ET-22-HK groups both showed similarly reduced gene-expression-levels of *ComDE* (*p* < 0.01, Figure 4C). On the contrary, the ET-22-S group showed a remarkably increased gene-expression-level of *ComDE* (*p* < 0.01, Figure 4C).

### 2.5. The Bioactive Substances for Resisting Formations of S. mutans Biofilms in Heat-Killed Bacteria and Secretions of L. paracasei ET-22

In order to investigate the potential bioactive substances that resist the formation of *S. mutans* biofilms, we analyzed the heat-killed bacteria and secretions of *L. paracasei* ET-22 in detail, using non-targeted metabonomics, whereby LC-MS/MS analysis was performed. This is a novel approach to confirming functional substances. Non-targeted metabonomics involves the comprehensive and systematic analysis of organic substances. An unbiased metabolomics analysis can contribute to the discovery of potential biomarkers. LC-MS/MS can provide more debris information that is needed for qualitative analysis, and it can reduce background noise. In addition, LC-MS/MS ensures that the spectrogram of trace components is free from the interference of abundant substances, which greatly improves the sensitivity of the detection. Therefore, in the present research, LC-MS/MS analysis is a more advantageous method of searching for possible functional substances in heat-killed bacteria and secretions of *L. paracasei* ET-22. As the results show, large amounts of organic substances were identified, including amines, nucleotides, organic acids, lipids, peptides, amino acids, terpene, and others in heat-killed bacteria (ET-22-HK) and secretions (ET-22-S) of *L. paracasei* ET-22. Of these, the nucleotides contained the most types of substances: more than 200 in ET-22-HK and 100 in ET-22-S (Figure 5A). Based on the relative ratio of peak areas, lipids represented the highest percentage, accounting for more than 20% in ET-22-HK and 10% in ET-22-S (Figure 5B). In turn, the contents of nucleotides, peptides, organic acids, and amines were relatively low, but exceeded 10% (Figure 5B). The amino acids, terpenes, and others accounted for less than 10% in ET-22-HK (Figure 5B). In each type of organic substance from ET-22-S, the top 10 types of substances made up more than 60% in the peak areas (Figure 5C). In ET-22-HK, the top 10 types of substance in each kind of organic substance accounted for more than 70% (Figure 5D).

Based on the debris information, the specific substances need to be further identified. By checking against the HMDB database (http://www.hmdb.ca/ (accessed on 3 January 2023)), Metlin (https://metlin.scripps.edu/ (accessed on 3 January 2023)), and the Majorbio cloud platform (https://cloud.majorbio.com (accessed on 3 January 2023)), the top 10 types of substances from ET-22-HK classified as organic acids, nucleotides, terpenes, peptides, amino acids, lipids, and amines were identified. The metabolite names, formulas, and relative ratio of peak areas of these substances are shown in Table 1. Similarly, the top 10 types of substances in each kind of the above organic substances from ET-22-S are listed in Table 2. For each kind of organic substance, some of the substances in the top 10 were the same for ET-22-HK and for ET-22-S. In the organic acids, 9,10-Epoxy-18-hydroxy-octadecanoic acid, D-2-Hydroxyglutaric acid, 3-Aminopentanedioic acid, 6-Hydroxyhexanoic acid, and phenyllactic acid all existed in abundance in ET-22-HK and ET-22-S (Table 1 and Table 2). ET-22-HK and ET-22-S also contained L-Lysine, L-Tyrosine, and L-Leucine (Table 1 and Table 2). In the lipids, most substances that included PG(i-12:0/i-19:0), PA(20:4(8Z,11Z,14Z,17Z)/PGF1alpha), SM(d16:2(4E,8Z)/20:5(6E,8Z,11Z,14Z,17Z)-OH(5)), PA(TXB2/20:0), PA(PGE1/20:1(11Z)), PE(20:5(5Z,8Z,11Z,14Z,17Z)/18:2(9Z,12Z)), PI(18:0/18:2(9Z,12Z)), PA(TXB2/22:1(13Z)) and PG(a-21:0/20:4(7E,9E,11Z,13E)-3OH(5S,6R,15S)) in the top 10 coexisted in ET-22-HK and ET-22-S (Table 1 and Table 2). In the amines, N1,N12-Diacetylspermine, N-Acetylcadaverine, N-[4-[Acetyl(3-aminopropyl)amino]butyl]-N-(3-aminopropyl)acetamide, N1,N8-Diacetylspermidine, N-Cyclohexylformamide, and 2-Phenylacetamide were all abundant in ET-22-HK and ET-22-S (Table 1 and Table 2). However, only hypoxanthine and xanthine were common in the nucleotides (Table 1 and Table 2). Moreover, there was no common substance in the top 10 of terpenes and peptides between ET-22-HK and ET-22-S (Table 1 and Table 2).

Other substances in the top 10 of organic acids, nucleotides, amino acids, lipids, and amines were all characteristic. For the organic acids, 9-Hydroxylinoleic acid, 2-Hydroxyadipic acid, DL-2-hydroxy stearic acid, behenic acid, and phytanic acid were peculiarly presented in ET-22-HK (Table 1). The 3-(4-Hydroxyphenyl)lactate, citramalic acid, acexamic acid, 12-hydroxyheptadecanoic acid, and azelaic acid were characteristic in ET-22-S (Table 2). In the nucleotides, ET-22-HK possessed abundant adenosine 3′-monophosphate, ADP-ribose, pseudouridine 5′-phosphate, adenosine monophosphate, guanidylic acid (guanosine monophosphate), zidovudine monophosphate, vidarabine, and 5′-Methylthioadenosine (Table 1). Notably, ET-22-S contained adenine, deoxyribose, FAPy-adenine, 7-Methylguanine, morph, oxypurinol, 5-Methyldeoxycytidine, and 1-(2-Hydroxyethyloxymethyl)-6-phenyl thiothymine (Table 2). In the amino acids, L-Carnitine, L-4-Hydroxyglutamate semialdehyde, citrulline, L-prolinamide, L-Glutamic Acid, cholylmethionine and N-Undecanoylglycine were particularly abundant in ET-22-HK (Table 1). ET-22-S contained abundant phenyl-Alanine, homocysteine, gamma-Glutamylvaline, tyrosine lactate, L-Phenylalanine, N-Acetyl-DL-Leucine, and N-Acetyl-DL-Phenylalanine (Table 2).

## 3. Discussion

### 3.1. Live L. paracasei ET-22 and Its Postbiotics Inhibit the Formation of S. mutans Biofilms

A biofilm is a collective lifestyle of microorganisms, and has many emergent characteristics, including interface aggregation, surface adhesion, water retention, nutrient absorption and retention, enhanced resistance, and collective cooperation. Once formed, the bacterial biofilm becomes a shelter for bacteria, which significantly enhances the resistance of bacteria to antimicrobial agents and their ability to escape from the host immune-system, leading to refractory and recurrent infections. In the oral cavity, caries is caused by oral biofilms formed by pathogenic *S. mutans* on the initial proteinaceous coating of the dental matrix [17]. Therefore, inhibiting the formation of the bacterial biofilms that cause dental caries, especially those of *S. mutans*, is important for the health of teeth. In view of the negative effects of antimicrobial agents, the biocontrol method has been recognized as a promising substitute. To verify whether the *L. paracasei* ET-22 can prevent the occurrence of dental caries in the oral cavity, the present research assessed the effects of *L. paracasei* ET-22 on the *S. mutans* biofilms, by co-culturing with *S. mutans*. To simulate the oral environment, the whole experiment used an artificial-saliva medium supplemented into 1% sucrose to culture *S. mutans*. After a 24 h co-culture, the live bacteria and secretions of *L. paracasei* ET-22 clearly showed significant inhibitory effects on the formation of *S. mutans* biofilms. This indicates that live *L. paracasei* ET-22 can inhibit the formation of *S. mutans* biofilms, by secreting metabolites. An unexpected phenomenon that emerged was that heat-killed *L. paracasei* ET-22 also showed a strong inhibition-effect on the formation of *S. mutans* biofilms, which had rarely been proved in previous studies. According to recently published research, postbiotics, productions from heat-killed probiotics, may play a stable and efficient role in protecting the health of the digestive tract [18]. Due to their colonization efficiency and stress resistance, live probiotics are less effective than postbiotics in some species such as *Akkermansia muciniphila*, which is used in the treatment of metabolic diseases [16]. Therefore, it can be inferred that the postbiotics of *L. paracasei* ET-22 may also inhibit the formation of *S. mutans* biofilms to protect patients from dental caries.

### 3.2. Live L. paracasei ET-22 and Its Postbiotics Inhibit the Formation of S. mutans Biofilms by Blocking the Initial Adhesion

EPS are the main components of oral biofilms [19,20]. By connecting with extracellular proteins, eDNA, and lipids, together they form biofilm matrices, which are conducive to bacterial colonization, biofilm formation and maintenance, and pathogenic realization [20]. *S. mutans* encodes several surface-associated GBP and glucosyltransferases, which serve as an integrated scaffold for biofilm formation by promoting the accumulation of local bacteria and forming a polymeric matrix [4,8,21]. To research the mechanisms whereby *L. paracasei* ET-22 inhibits the formation of *S. mutans* biofilms, we evaluated the effects of live bacteria, heat-killed bacteria, and secretions of *L. paracasei* ET-22 on the gene expressions of *brpA*, *gtfB*, *gbpB, gbpC*, and *gbpD* of *S. mutans*. Among them, the water-insoluble glucan synthesized by gtfB, a glucosyltransferase of the GTF cluster encoded by the *gtfB* gene, was identified as the main component of EPS in *S. mutans* biofilms. In addition, brpA is involved in the biofilm formation and stress tolerance of *S. mutans*. The mutant strain lacking the *brpA* gene has a limited ability to grow and accumulate on solid surfaces [21]. Interestingly, our findings revealed that live *L. paracasei* ET-22 and its postbiotics not only reduced the gene expression of *gtfB*, but also reduced the gene-expression levels of *gbpB-D* in the GBP cluster and *brpA*. Similarly, previous studies have shown that *Lactobacillus* decreases the gene expression of *brpA*, the GBP cluster, and glucosyltransferase genes of the GTF cluster in *S. mutans*. One study found reduced biofilm-formation and lower gene-expression of *gbpB* and *gtfB* in *S. mutan*s after co-culture with *L. casei* [22]. Another study also observed that, after a coculture with gellan containing *L. paracasei* 28.4, the *S. mutans* showed decreased biofilm-formation, accompanied by reduced gene-expression of *brpA* and *gtfB*-*D* [23]. These data suggest that live *L. paracasei* ET-22 and its postbiotics may inhibit the production of EPS by inhibiting the expression of the GTF cluster and brpA to defend against the formation of *S. mutans* biofilms. Secretions of *L. paracasei* ET-22 also showed inhibitory effects on the gene expression of *brpA*, *gtfB*, *gbpB, gbpC*, and *gbpD* of *S. mutans*, thereby blocking the construction of the biofilm structure. This indicates that suppressing the production of EPS by secretions is one of the pathways by which live *L. paracasei* ET-22 limits the formation of *S. mutans* biofilms. In line with our results, other studies also found that secretions of probiotics mainly change the biofilm structure by changing the expression of virulence genes related to the formation of *S. mutans* biofilms [17].

Adhesion is a prerequisite for the formation of biofilms. We aimed to determine whether *L. paracasei* ET-22 could affect the adhesion of *S. mutans.* The initial adhesion of *S. mutans* is achieved when it combines with the membrane receptors on the tooth surface via sucrose-independent weak interactions, which are mediated by long-range forces, i.e., van der Waals forces and Coulomb interactions, and short-range forces, including electrostatic interactions and hydrophobic interactions [24,25,26]. SPAP, a non-sucrose-dependent adhesion protein, is initially and primarily involved in promoting the colonization of *S. mutans* on tooth surfaces by specific interactions with salivary agglutinins [27]. Some studies have shown that SPAP-deficient *S. mutans* mutants showed a decreased ability to bind to salivary components and the cell matrix, as well as decreased co-aggregation-activity [28]. In this study, the *SpaP* expression of *S. mutans* was significantly downregulated by live *L. paracasei* ET-22 and its postbiotics. Meanwhile, they both reduced the initial adhesion-levels of *S. mutans* to hydroxyapatite. Consistently with these results, the CSLM results also showed the inhibitory effects of live *L. paracasei* ET-22 and its postbiotics on the formation of *S. mutan*s biofilms. Therefore, it can be speculated that *L. paracasei* ET-22 and its postbiotics may inhibit the initial adhesion of *S. mutans* by changing the weak interactions between *S. mutans* and the tooth surface. A previous report found that postbiotics of *L. paracasei* may change the weak interactions during the initial adhesion of *S. mutans* [29]. This is in line with the findings of this study.

### 3.3. Live L. paracasei ET-22 and Its Postbiotics Inhibit S. mutans Biofilms by Interfering with the QS System and the Expression of Virulence Factors

Acid production and acid tolerance are considered the main virulence characteristics of *S. mutans* [3]. The inhibition of acid production and the regulation of acid sensitivity of *S. mutans* are important means of preventing dental caries [6,30,31,32]. *LDH* is an important acid-producing gene of *S. mutans* involved in the process of carbohydrate metabolism. It plays an important role in the process of tooth demineralization [13]. Some proteins, such as *recA* and *ffh*, could induce the DNA-repair mechanism of *S. mutans* to enhance the acid-tolerance capacity [33]. In addition, *relA* can improve tolerance of *S. mutans* to environmental stress, through the synthesis of (p)ppGpp [34]. Our study found that the transcriptions of *LDH*, *ffh*, *recA*, and *relA* were downregulated by *L. paracasei* ET-22 and its postbiotics, which may contribute to the inhibition of the formation of *S. mutans* biofilms by decreasing its acid-producing and acid-tolerance functions. Similarly, another study showed that the presence of *L. rhamnosus* significantly reduced the expression of LDH and then effectively modulated the formation of cariogenic biofilms [35]. To sum up, this evidence suggests that live *L. paracasei* ET-22 and its postbiotics may inhibit the formation of *S. mutans* biofilms by decreasing acid-production and acid-tolerance functions.

In the process of biofilm formation, various bacterial species communicate with others through the QS system in biofilms [3]. The ComCDE system is the most common intraspecific communication approach of QS systems in *S. mutans* biofilms, in which *ComDE* and *comX* can regulate the communications of *S. mutans* by competitive signaling-peptides [36]. The inactivation of the ComDE pathway may result in defective *S. mutans* biofilms [36,37]. The reduction in the gene expression of *ComDE* suppresses the expression of cariogenic virulence-factors in *S. mutans* [3,36]. Suppressed expression of *ComDE* and cariogenic *LDH*, *ffh*, *recA*, and *relA* genes indicate that live *L. paracasei* ET-22 and its postbiotics inhibit the expression of cariogenic virulence-factors by suppressing the ComDE pathway. However, the secretions of *L. paracasei* ET-22 showed the opposite effect on the gene expression of *ComDE*. The reason for this is unclear, and requires further research. Meanwhile, this finding also suggests that the bodies, but not the secretions, of *L. paracasei* ET-22 interfere with the QS system

### 3.4. The Inhibitory Effect of Live L. paracasei ET-22 and Its Postbiotics on S. mutans Biofilms Is Mediated by Multiple Components

Further research was required to determine which specific substances interfere with the QS system. Non-targeted LC-MS/MS analysis was used to further investigate the bioactive substances of the bodies and postbiotics of *L. paracasei* ET-22. Large numbers of organic substances were identified. Multiple organic acids can inhibit the formation of bacterial biofilms. On the one hand, an acidic environment may be detrimental for the survival and physiological activity of pathogens. On the other hand, some organic acids can inhibit the formation of biofilms through specific mechanisms. In the common organic acids of the postbiotics and secretions of *L. paracasei* ET-22, abundant phenyllactic acid was reported to show broad-spectrum inhibition of the biofilm formation of pathogenic bacteria, such as *Actinobacillus actinomycetemcomitans*, *Listeria monocytogenes*, *Staphylococcus aureus*, and *Salmonella enteritidis* [38]. Several studies have found that the suppressed effects were all involved in the transcriptional inhibitions of genes controlling the syntheses and secretions of exopolysaccharides, extracellular proteins, and virulence factors [39,40,41,42]. Another study further confirmed that phenyllactic acid played an inhibitive role on the QS system, including the Rhl and PQS QS systems, to suppress the gene expression of relative virulence-factors and biofilm development [43]. In agreement with previous findings, the live *L. paracasei* ET-22 and its postbiotics may inhibit the ComDE QS pathway in restraining the *S. mutans* biofilms partly by phenyllactic acids. In the characteristic nucleotides of postbiotics from *L. paracasei* ET-22, zidovudine has been reported to show strong activity against the biofilms of *S. Typhimurium* and *E. coli* [44]. Therefore, zidovudine monophosphate may also be a functional substance in the postbiotics of *L. paracasei* ET-22 that limits the development of *S. mutans* biofilms. However, the regulated pathway is unclear. Citrullination mediated by PPAD can constrain the formation of oral *P. gingivalis* biofilms, which is involved in the occurrence of periodontitis and other oral diseases [45]. In this process, citrulline is used to modify the arginine residues of gingipain-derived adhesin proteins [45]. Similarly, abundant citrulline in postbiotics of *L. paracasei* ET-22 may also be involved in inhibiting the development of *S. mutans* biofilms by the PPAD-citrullination pathway, as shown in the present study. Unlike the direct effects of the above functional substances, the characteristic metabolite N-undecanoylglycine can inhibit pathogenic bacteria and their physiological activities, through host immunity. Bacterial N-undecanoylglycine can be sensed by epithelial cells and activate the antimicrobial immunity of mucosa, such as in intestinal Tuft-2 cells [46]. Therefore, N-undecanoylglycine in the postbiotics of *L. paracasei* ET-22 may stimulate the oral mucosa to inhibit the development of *S. mutans* and its biofilms. However, the effects of these potent functional substances still need further confirmation. Additionally, there are still many common and specific substances in the postbiotics and secretions of *L. paracasei* ET-22, which may contain another substance that inhibits *S. mutans* biofilms or interferes with the QS system. The current research evidence is still insufficient. It should be noted that the research on postbiotics of *L. paracasei* ET-22 and its active substances still need further confirmation through animal experiments.

Dental caries is a result of biofilms formed on the teeth surface by *S. mutans*. Different *Lactobacillus* strains have different effects on the formation of *S. mutans* biofilms [47]. This study found that live bacteria, heat-killed bacteria, the ComDE QS pathway and secretions of *L. paracasei* ET-22 all inhibited the formation of *S. mutans* biofilms. They changed the weak-interaction forces, initial adhesions, and structure of *S. mutans* biofilms by competing with salivary membranes, reducing the expression of adhesion proteins, EPS, and virulence factors, and interfering with the ComDE QS pathway. The results from the present study provide new evidence for the potential efficacy of postbiotics in the prevention of dental caries. The postbiotics of *L. paracasei* ET-22 have strong functional activity as live bacteria, and might broaden the application of *L. paracasei* ET-22 for oral health. According to our results, considering the colonization efficiency of live bacteria, postbiotics of *L. paracasei* ET-22 may be a more stable product for use in protection against dental decay induced by *S. mutans*. Moreover, the application of postbiotics derived from *L. paracasei* ET-22 could reduce the potential risks brought about by live bacteria. Therefore, postbiotics of *L. paracasei* may be candidates for the prevention of dental caries.

## 4. Materials and Methods

### 4.1. Bacterial Strains and Culture Medium

The *L. paracasei* ET-22 was obtained from the Sanhe Fucheng Biotechnology Co. Ltd. (Sanhe, China). The *S. mutans* (CICC 10387) was acquired from the China Center of Industrial Culture Collection (Beijing, China). *L. paracasei* and *S. mutans* were cultured with commercial deMan, MRS, and BHI mediums (Beijing Land Bridge Technology Co., Beijing, China), at 37 °C. Before the experiments, all strains were stored at −80 °C.

### 4.2. Preparation of Live Bacteria, Heat-Killed Bacteria, and Secretions of L. paracasei ET-22

*L. paracasei* ET-22 was cultured overnight and then centrifuged at 10,000× *g* for 10 min. After being washed with PBS three times, the bacterial solution was resuspended and adjusted to 5.5 × 10^8^ CFU/mL with PBS with a spectrophotometer value at OD600. After the culture on MRS agar medium (Beijing Land Bridge Technology Co., China), the bacterial concentration was further confirmed by viability count. Using this method, the live *L. paracasei* ET-22 was prepared. Using a similar procedure, bacterial solutions adjusted to 5.5 × 10^8^ CFU/mL were inactivated by a water bath at 70 °C for 1 h, following the method used in a previous study [48]. Then, the heat-killed *L. paracasei* ET-22 was acquired. According to a previously reported method, the live bacteria adjusted to 5.5 × 10^8^ CFU/mL were washed and resuspended with PBS (bacterial sludge (v):PBS (v) = 1:3) [49]. After being stirred at 60 g at room temperature for 2 h and then centrifuged at 10,000× *g* for 10 min, the supernatant was filtered with the 0.22 μm sterile filter membranes. The filtrate comprised the secretions of *L. paracasei* ET-22. The live bacteria, heat-killed bacteria, and secretions of *L. paracasei* ET-22 were stored at 4 °C.

### 4.3. Treatment for the Formation of S. mutans Biofilms

First, 1.9 mL artificial-saliva medium (Solarbio, Beijing, China) supplemented with 1% sucrose and hydroxyapatite discs (Clarkson Chromatography Products, Inc., South Williamsport, PA, USA) was added into sterile 24-well polystyrene culture plates (Solarbio, China) [50]. The hydroxyapatite discs were used to support the biofilm formation. The *S. mutans* were inoculated into the culture system to reach the 5 × 10^8^ CFU/mL. To investigate the effects of *L. paracasei* ET-22 on the formation of *S. mutans* biofilms, we used live bacteria, heat-killed bacteria, and secretions of *L. paracasei* ET-22 to coculture with *S. mutans*. As Table 1 shows, a single *S. mutans* culture (Model group) was used as a negative control. The live bacteria (ET-22-L), heat-killed bacteria (ET-22-HK), and secretions (ET-22-S) of *L. paracasei* ET-22 were used to coculture with *S. mutans*. In each group, 50 μL PBS or live bacteria, heat-killed bacteria, and secretions of *L. paracasei* ET-22 were added into the culture system to coculture with *S. mutans*. After anaerobic culture for 24 h, the formation of *S. mutans* biofilms was assessed. The anaerobic-culture process was finished in anaerobic-culture bags (Hopebio, Yancheng, China), and the residual oxygen inside the bag was cleared, using an AnaeroPack (Mitsubishi Gas Chemical Company, Inc., Tokyo, Japan). Group setting and culture conditions are as described in Table 3.

To assess the effects of the above treatments on the formation of *S. mutans* biofilms, the formation levels were analyzed using the crystal-violet staining as previously described [51]. In this method, the culture medium was cleared away. Then the biofilms were washed with PBS. After finishing the staining, the results were read with ELIASA (Perkin Elmer, PE Victor X3, Waltham, MA, USA) at Abs595nm.

### 4.4. Extracellular-Polysaccharide Production in Biofilms

The extracellular polysaccharide (EPS) productions of the *S. mutans* biofilms were measured using a previously described method [19,20]. Briefly, the resuspended biofilms were centrifuged in ultrapure water at 10,000× *g* for 10 min. After repeating three times, the collected supernatants were transferred into a new centrifuge tube with three times the volume of absolute ethanol. The solutions were mixed and then placed at −20 °C for 30 min to precipitate soluble polysaccharides. After centrifugation by 10,000× *g* for 10 min and washing with 70% alcohol, the soluble polysaccharides were resuspended and resolved with 5 mL NaOH solution at 1 M. Subsequently, 2 mL 5% phenol (*m*:*v*) and 5 mL of concentrated sulfuric acid were added, and the samples were treated away from light at room temperature for 10 min. The dissolved substances were soluble EPS. Having been vortexed for 30 s and placed into a water bath at 25 °C for 20 min, the samples’ absorbance was read at 490 nm to determine the soluble EPS contents.

After being repeatedly dissolved, the precipitates from the biofilms in ultrapure water were resuspended with 5 mL NaOH solution at 1 M and then shaken for 15 min. Subsequently, the solutions were centrifuged at 10,000× *g* for 10 min. The dissolved substances were insoluble EPS. The remainder of the operation was the same as the method used for soluble-EPS determination. After the process of precipitation, dissolution, reaction with phenol and sulfuric acid, a water bath, and reading values at 490 nm, the contents of insoluble EPS were acquired.

### 4.5. Biofilm Microstructure Observed Using Scanning Electron Microscopy

Scanning electron microscopy (SEM) was applied to observe the structure of biofilms as previously described [52]. Based on the culture method of *S. mutans* biofilms, glass slides were placed below the hydroxyapatite discs in the wells of 24-well polystyrene culture plates. After anaerobic culture for 24 h, the liquid culture medium was gently discarded, and PBS was used to wash the *S. mutans* biofilms 3 times. The *S. mutans* biofilms on glass slides were fixed at 4 °C overnight with PBS containing 2.5% glutaraldehyde. Glass slides were taken out and dehydrated with gradient-increased alcohol solutions from 30% (*v*:*v*) to 100% (*v*:*v*). Subsequently, the glass slides were dried in an oven at 37 °C for 24 h. After being sprayed gold, the *S. mutans* biofilms were observed with S-4800 SEM (HITACHI, Tokyo, Japan).

### 4.6. Detection of Biofilm Thickness with Confocal Laser Scanning Microscopy

Using the same method described in 4.5, *S. mutans* biofilms formed on hydroxyapatite discs. Live bacteria and dead bacteria in the biofilms were labeled with 2% N01 (*m*:*v*) (BBcellProbe^®^, Shanghai, China) and 5% PI (*m*:*v*) (BBcellProbe^®^, China), respectively [53]. After being washed with normal saline three times, the biofilms were stained with 500 μL 2% N01 in PBS and incubated at 25 °C for 15 min. Subsequently, the biofilms were washed twice with normal saline. Then, 500 μL 5% PI was added to incubate the biofilms at 25 °C for 15 min. Confocal imaging was performed using an LSM900 instrument (Leica, Wetzlar, Germany), in which the total magnification was 200× and the magnification of the oil lens was 630×. During the image acquisition, >600 nm was set for PI and 488–525 nm was used for N01. Moreover, the thickness of the biofilm was determined, using a 5 μm thick (8-bit, 1024 × 1024 pixels) Z-slice method. The biovolume of the biofilm and the layer-coverage distribution were quantified from the confocal stacks, using the image-processing software COMSTAT [54].

### 4.7. RT-qPCR for Gene-Expression Levels

After finishing the treatment, the RNAs were extracted and purified using the optimized protocol described by Kuhnert et al. [28]. Extracted crude RNAs were enzyme-catalyzed to remove contaminated genomic DNAs. Agarose gel electrophoresis was used to verify the integrity of the RNAs. Purified RNAs were reverse transcribed to cDNAs, using the PrimeScript^TM^ 1st Strand cDNA Synthesis Kits (Takara, Kusatsu, Japan). A quantitative real-time reverse-transcription polymerase-chain-reaction (qRT-PCR) was performed using the Applied Biosystems StepOne^TM^ system (Thermo Fisher, Waltham, MA, USA), in which the PowerUp^TM^ SYBR^TM^ Green Master Mix (Thermo Fisher, USA) was used. Primers for target genes were designed for initial adhesion-protein production, glucan production, glucan-protein production, acid tolerance, and quorum sensing in *S. mutans*. The sequences are shown in Table 4. Primers were synthesized by Tsingke Biotechnology Co. Ltd. (Beijing, China). PCR reactions were performed as previously described [55]. The cycling conditions were as follows: one cycle at 95 °C for 5 min; 40 cycles for denaturation at 95 °C for 30 s, annealing at 52–58 °C (depending on the primers used) for 30 s; and extension- and fluorescent-data collection at 72 °C for 30 s. Dissociation curves were generated, followed by the cycling reactions. The expressions levels of the target genes were normalized to 16S rRNA [56]. Data were calculated using the 2^−ΔΔCT^ method [55].

### 4.8. Active-Substances Determinations in Heat-Killed L. paracasei ET-22 by Non-Targeted Metabonomics

Here, 50 mg solid heat-killed *L. paracasei* ET-22 dried by freezing were accurately weighed, and the organic substances were extracted using a 400 µL 80% methanol aqueous solution (*v*:*v*) supplemented with 0.02 mg/mL L-2-chlorophenylalanin as an internal standard. The mixture was processed using a tissue crusher Wonbio-96c (Shanghai Wanbo Biotechnology Co., Ltd., Shanghai, China) at 50 Hz for 6 min at −10 °C followed by ultrasound treatment at 40 kHz for 30 min, at 5 °C. After being placed at −20 °C for 30 min to precipitate proteins, and a centrifugation by 13,000× *g* at 4 °C for 15 min, the supernatant was transferred for non-targeted LC-MS/MS analysis. A pooled quality-control sample was prepared by mixing equal volumes of all samples. The quality-control and analytic samples were tested in the same manner. Briefly, samples were separated by an HSS T3 column (100 mm × 2.1 mm, i.d., 1.8 μm) and then entered for mass-spectrometry detection. The mobile phases consisted of a 0.1% formic-acid aqueous solution (*v*:*v*): acetonitrile = 95:5 (*v*:*v*, solvent A) and 0.1% formic acid in acetonitrile (*v*:*v*): isopropanol: water = 47.5:47.5: 5 (*v*:*v*, solvent B). The injection volume of the sample was 2 µL and the flow rate was set to 0.4 mL/min. The column temperature was maintained at 40 °C. During the period of analysis, all of these samples were stored at 4 °C. The mass spectrometric data were collected using a Thermo UHPLC-Q Exactive HF-X Mass Spectrometer (Thermo Scientific high-resolution) equipped with an electrospray-ionization source operating in either the positive- or negative-ion mode. The optimal conditions were set as follows: heater temperature at 425 °C; capillary temperature at 325 °C; sheath-gas flow rate at 50 arb; aux-gas flow rate at 13 arb; ion-spray voltage floating (ISVF), −3500 V in the negative mode and 3500 V in the positive mode; normalized collision energy, 20–40–60 V rolling for MS/MS. The full MS-resolution was 60,000, and the MS/MS resolution was 7500. Data acquisition was performed using the data-dependent acquisition (DDA) mode. The detection was carried out over a mass range of 70–1050 *m*/*z*. The raw data of LC-MS/MS were preprocessed using the Progenesis QI software (Waters Corporation, Milford, MA, USA), and a three-dimensional data matrix in CSV format was exported. The information in this three-dimensional matrix includes sample information, metabolite names, and mass-spectral-response intensities. Internal-standard peaks, as well as any known false positive-peaks including noise, column bleed, and derivatized-reagent peaks, were removed from the data matrix. Effect peaks were pooled. At the same time, the metabolites were searched and identified using the HMDB database (http://www.hmdb.ca/ (accessed on 3 January 2023)) and Metlin (https://metlin.scripps.edu/ (accessed on 3 January 2023)).

The data were further uploaded to the Majorbio cloud platform (https://cloud.majorbio.com (accessed on 3 January 2023)) for data analysis. At least 80% of the metabolite features detected in any sample were retained. After being filtered, the minimum metabolite-values were imputed for specific samples, in which the metabolite levels fell below the low limit of quantitation and each metabolic feature was normalized by sum. In order to reduce the errors caused by sample preparation and instrument instability, the response intensities of the samples’ mass-spectrum peaks were normalized using the sum-normalization method, and the normalized-data matrix was obtained. At the same time, the variables with relative standard deviation (RSD) >30% of the quality-control sample were removed, and log10 transformation was performed to obtain the final data matrix for subsequent analysis.

### 4.9. Statistics

All data are shown as the mean ± SEM. Experimental results were analyzed using one-way analysis of variance (ANOVA) with an additional Dunnett’s test in GraphPad Prism 8 software (San Diego, CA, USA). Results from the confocal laser scanning microscopy were analyzed using Image J and Contest 2 (Technical University of Denmark, Lyngby, Denmark). When *p* < 0.05 and *p* < 0.01, the difference was significant and extremely significant, respectively. * *p* < 0.05, ** *p* < 0.01.

## 5. Conclusions

Our work suggests that live *L. paracasei* ET-22 and its postbiotics and secretions display inhibition effects on the adhesion and formation of *S. mutans* biofilms. Postbiotics of *L. paracasei* ET-22 have great potential as a biological anticariogenic-agent that acts against *S. mutans* biofilms to prevent dental decay.

## Figures and Tables

**Figure 1 molecules-28-01236-f001:**
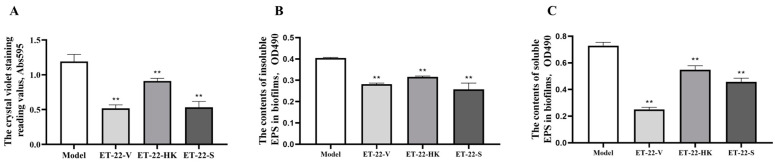
Effects of *L. paracasei* ET-22 on formation of *S. mutans* biofilms. (**A**). The effects of live bacteria (ET-22-V), heat-killed bacteria (ET-22-HK), and secretions (ET-22-S) of *L. paracasei* ET-22 on the formation levels of *S. mutans* biofilms. The single pathogen *S. mutans* (Model) was cultured as a control. Means ± SEM are shown (*n* = 6). ** *p* < 0.01, vs. Model group. (**B**). The effects of live bacteria, heat-killed bacteria, and secretions of *L. paracasei* ET-22 on the contents of insoluble extracellular polysaccharides (EPS) in *S. mutans* biofilms. Means ± SEM are shown (*n* = 6). ** *p* < 0.01, vs. Model group. (**C**). The effects of live bacteria, heat-killed bacteria, and secretions of *L. paracasei* ET-22 on the contents of soluble EPS in *S. mutans* biofilms. Means ± SEM are shown (*n* = 6). ** *p* < 0.01, vs. Model group.

**Figure 2 molecules-28-01236-f002:**
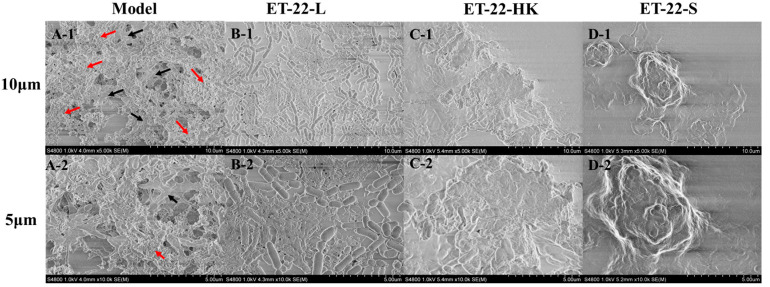
Biofilm microstructure of *S. mutans* after the co-culture with live bacteria (ET-22-V), heat-killed bacteria (ET-22-HK), and secretions (ET-22-S) of *L. paracasei* ET-22, produced using scanning electron microscopy. (**A-1**,**A-2**) are typical images of *S. mutans* biofilms in the Model group under a 10 μm (**A-1**) or 5 μm (**A-2**) scale. The red arrow points to the “corolla” of the cauliflower-like structure. The black arrow points to the “flower stalk” of the cauliflower-like structure. (**B-1**,**B-2**) are representative images of *S. mutans* biofilms after the co-culture with live *L. paracasei* ET-22 under a 10 μm (**B-1**) or 5 μm (**B-2**) scale. (**C-1**,**C-2**) are representative images of *S. mutans* biofilms after the co-treatment with heat-killed *L. paracasei* ET-22 under a 10 μm (**C-1**) or 5 μm (**C-2**) scale. (**D-1**,**D-2**) are representative images of *S. mutans* biofilms after the co-treatment with secretions of *L. paracasei* ET-22 under a 10 μm (**D-1**) or 5 μm (**D-2**) scale.

**Figure 3 molecules-28-01236-f003:**
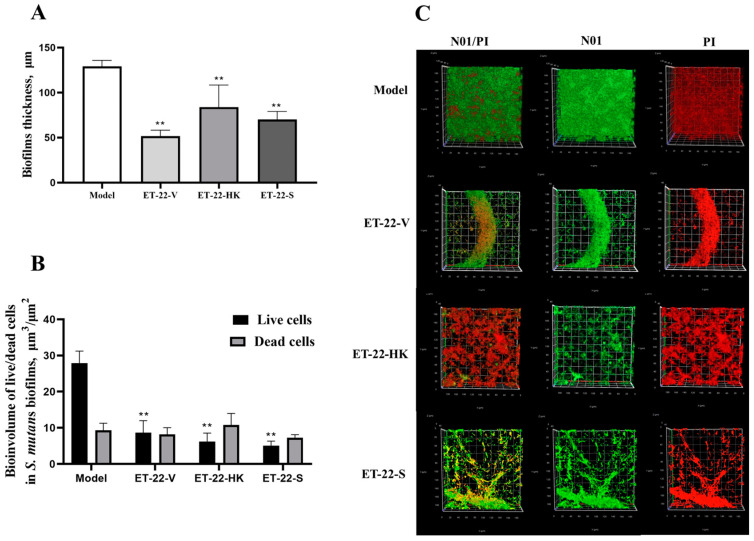
Effects of *L. paracasei* ET-22 on the thickness and biomasses of *S. mutans* biofilms. (**A**) Changes in thickness of the *S. mutans* biofilms after the co-culture with live bacteria (ET-22-V), heat-killed bacteria (ET-22-HK), and secretions (ET-22-S) of *L. paracasei* ET-22. Means ± SEM are shown (*n* = 6). ** *p* < 0.01, vs. Model group. (**B**) The staining results of live and dead bacteria in *S. mutans* biofilms viewed using confocal laser scanning microscopy. (**C**) The biomasses of live bacteria (green) and dead bacteria (red) in *S. mutans* biofilms of the ET-22-V, ET-22-HK, and ET-22-S groups. N01 was used for the nuclear staining of live bacteria. Propidium iodide (PI) was used for the nuclear staining of dead bacteria. Means ± SEM are shown (*n* = 6). ** *p* < 0.01, vs. live bacteria in the Model group. (**D**,**E**), the gathering positions of living bacteria (**D**) and dead bacteria (**E**) above the bottom of hydroxyapatite discs.

**Figure 4 molecules-28-01236-f004:**
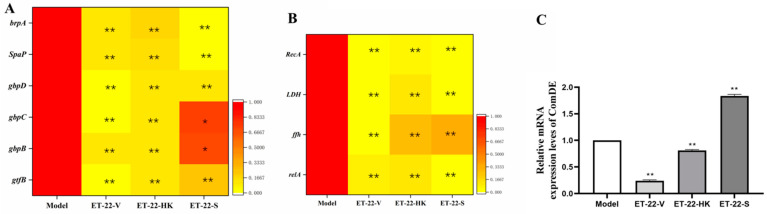
Effects of *L. paracasei* ET-22 on gene expressions involved in the adhesive, acid-resistant, acidogenic, and quorum-sensing (QS) qualities of *S. mutans*. (**A**) A heatmap of the relative gene-expression-levels of proteins involving adhesions and the formation of the *S. mutans* biofilm, including *brpA*, *SpaP*, *gbpD*, *gbpC*, *gbpB*, and *gtfB*. Means ± SEM are shown (*n* = 6). * *p* < 0.05, ** *p* < 0.01, vs. Model group. (**B**) A heatmap of the relative gene-expression-levels of proteins involved in the acid production and acid tolerance of *S. mutans*, including *RecA*, *LDH*, *ffh*, and *relA*. Means ± SEM are shown (*n* = 6). ** *p* < 0.01, vs. Model group. (**C**) The gene-expression-levels of *comDE* in the QS system of *S. mutans*. Means ± SEM are shown (*n* = 6). ** *p* < 0.01, vs. Model group.

**Figure 5 molecules-28-01236-f005:**
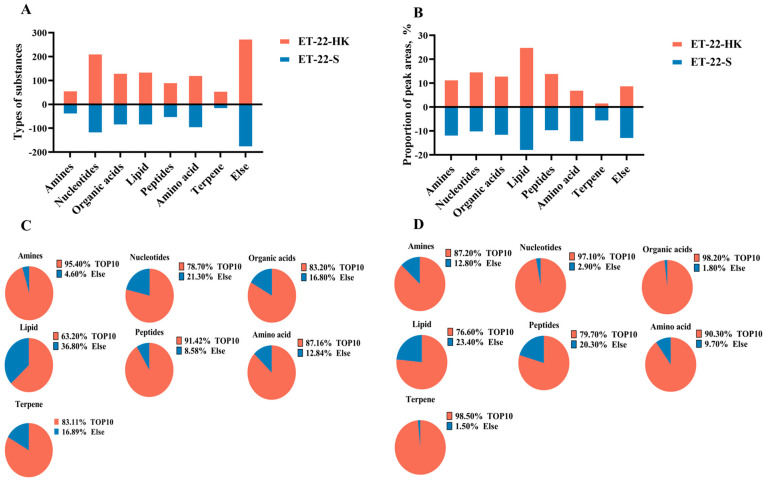
The main organic substances in heat-killed bacteria and secretions of *L. paracasei* ET-22. (**A**) The number of substance types in each kind of organic substance from heat-killed bacteria (ET-22-HK) and secretions (ET-22-S) of *L. paracasei* ET-22. (**B**) The proportion of relative peak areas for each kind of organic substance. (**C**) The proportion of peak areas for the top 10 substance types in each kind of organic substance from ET-22-HK. (**D**) The proportion of peak areas for the top 10 substance types in each kind of organic substance from ET-22-S.

**Table 1 molecules-28-01236-t001:** Major compositions in ET-22-HK by non-targeted metabonomics ^1^.

No.	Metabolite	Formula	Peak Area (%)	No.	Metabolite	Formula	Peak Area (%)
Organic acids				7	Arginylproline	C11H21N5O3	3.24
1	9,10-Epoxy-18-hydroxy-octadecanoic acid	C18H34O4	14.55	8	1-(D-3-Mercapto-2-methyl -1-oxopropyl)-L-proline	C9H15NO3S	3.01
2	9-Hydroxylinoleic acid	C18H32O3	13.12	9	L-beta-aspartyl-L-leucine	C10H18N2O5	2.93
3	6-Hydroxyhexanoic Acid	C6H12O3	12.23	10	Valylserine	C8H16N2O4	2.43
4	3-Aminopentanedioic Acid	C5H9NO4	9.48	11	Others	-	8.58
5	D-2-Hydroxyglutaric acid	C5H8O5	7.44	Amino acid			
6	2-Hydroxyadipic acid	C6H10O5	7.36	1	L-Leucine	C6H13NO2	32.23
7	DL-2-hydroxy stearic acid	C18H36O3	5.96	2	L-Lysine	C6H14N2O2	10.83
8	Behenic acid	C22H44O2	5.74	3	Citrulline	C6H13N3O3	9.73
9	Phytanic acid	C20H40O2	4.26	4	L-Carnitine	C7H15NO3	9.40
10	Phenyllactic acid	C9H10O3	3.06	5	L-Tyrosine	C9H11NO3	8.14
11	Others	-	16.80	6	L-prolinamide	C5H10N2O	4.41
Nucleotides				7	L-4-Hydroxyglutamate semialdehyde	C5H9NO4	3.92
1	Adenosine 3′-monophosphate	C10H14N5O7P	28.17	8	N-Undecanoylglycine	C13H25NO3	3.62
2	Hypoxanthine	C5H4N4O	15.92	9	Cholylmethionine	C29H49NO6S	3.23
3	5′-Methylthioadenosine	C11H15N5O3S	11.31	10	L-Glutamic Acid	C5H9NO4	1.65
4	Zidovudine monophosphate	C10H14N5O7P	3.83	11	Others	-	12.84
5	ADP-ribose	C15H23N5O14P2	3.56	Lipid			
6	Adenosine monophosphate	C10H14N5O7P	3.29	1	PA(PGE1/20:1(11Z))	C43H77O11P	9.35
7	Pseudouridine 5′-phosphate	C9H13N2O9P	3.18	2	SM(d16:2(4E,8Z)/20:5(6E, 8Z,11Z,14Z,17Z)-OH(5))	C41H71N2O7P	8.59
8	Guanidylic acid (guanosine monophosphate)	C10H14N5O8P	3.16	3	PA(TXB2/22:1(13Z))	C45H81O12P	8.03
9	Xanthine	C5H4N4O2	3.15	4	PI(18:0/18:2(9Z,12Z))	C45H83O13P	6.88
10	3′,5′-Cyclic AMP	C10H12N5O6P	3.13	5	PA(TXB2/20:0)	C43H79O12P	6.61
11	Others	-	21.30	6	PG(a-21:0/20:4(7E,9E, 11Z,13E)3OH(5S,6R,15S))	C47H85O13P	6.52
Terpene				7	PE(LTE4/20:1(11Z))	C48H85N2O11PS	6.33
1	Glycinoeclepin B	C31H42O9	16.02	8	PA(20:4(8Z,11Z,14Z,17Z)/PGF1alpha)	C43H73O11P	4.98
2	Soyasapogenol F	C30H50O3	14.99	9	PG(i-12:0/i-19:0)	C37H73O10P	3.36
3	Alpha-Campholonic acid	C10H16O3	10.71	10	PI(PGF1alpha/22:2(13Z,16Z))	C51H91O16P	2.55
4	Zedoarol	C15H18O3	9.55	11	Others	-	36.80
5	Geranylcitronellol	C20H36O	7.68	Amines			
6	Arctiopicrin	C19H26O6	5.88	1	N1,N12-Diacetylspermine	C14H30N4O2	54.59
7	Canavalioside	C26H42O12	5.28	2	N1,N8-Diacetylspermidine	C11H23N3O2	21.17
8	Manoalide	C25H36O5	5.21	3	Oleamide	C18H35NO	11.07
9	Cinncassiol C	C20H28O7	4.91	4	N-Palmitoyl Cysteine	C14H22O4	2.38
10	3-*O*-*cis*-Coumaroylmaslinic acid	C39H54O6	2.88	5	N1-Acetylspermine	C12H28N4O	1.55
11	Others	-	16.89	6	9-Octadecenamide	C18H35NO	1.27
Peptides				7	N-[4-[Acetyl(3-aminopropyl)amino]butyl]-N-(3-aminopropyl)acetamide	C14H30N4O2	1.21
1	Permetin A	C54H92N12O12	52.51	8	2-Phenylacetamide	C8H9NO	0.92
2	Prolyl-Asparagine	C9H15N3O4	5.82	9	N-Acetylcadaverine	C7H16N2O	0.64
3	Sarcodon scabrosus Depsipeptide	C23H39N3O8	6.44	10	Manumycin A	C31H38N2O7	0.60
4	Cyclo(his-pro)	C11H14N4O2	6.01	11	Others	-	4.60
5	Etelcalcetide	C38H73N21O10S2	4.91				
6	Asp-Phe	C13H16N2O5	4.12				

Note: ^1^ ET-22-HK, heat-killed *L. paracasei* ET-22.

**Table 2 molecules-28-01236-t002:** Major compositions in ET-22-S by non-targeted metabonomics ^1^.

No.	Metabolite	Formula	Peak Area (%)	No.	Metabolite	Formula	Peak Area (%)
Organic acids				10	Gamma-Glutamylmethionine	C10H18N2O5S	2.47
1	6-Hydroxyhexanoic Acid	C6H12O3	58.01	11	Others	-	20.30
2	Phenyllactic acid	C9H10O3	14.12	Amino acid			
3	3-Aminopentanedioic Acid	C5H9NO4	11.02	1	N-Acetyl-DL-Leucine	C8H15NO3	20.23
4	3-(4-Hydroxyphenyl)lactate	C9H10O4	7.03	2	N-Acetyl-DL-Phenylalanine	C11H13NO3	14.96
5	9,10-Epoxy-18-hydroxy-octadecanoic acid	C18H34O4	2.51	3	L-Leucine	C6H13NO2	10.88
6	D-2-Hydroxyglutaric acid	C5H8O5	2.00	4	Gamma-Glutamylvaline	C10H18N2O5	10.31
7	Citramalic Acid	C5H8O5	1.70	5	Phenyl-Alanine	C9H11NO2	9.47
8	acexamic acid	C8H15NO3	1.13	6	Homocysteine	C4H9NO2S	7.09
9	12-hydroxyheptadecanoic acid	C17H34O3	0.37	7	Tyrosine lactate	C12H15NO5	6.87
10	Azelaic Acid	C9H16O4	0.31	8	L-Phenylalanine	C9H11NO2	5.20
11	Others	-	1.80	9	L-Tyrosine	C9H11NO3	2.80
Nucleotides				10	L-Lysine	C6H14N2O2	2.49
1	Hypoxanthine	C5H4N4O	58.42	11	Others	-	9.70
2	FAPy-adenine	C5H7N5O	15.41	Lipid			
3	Xanthine	C5H4N4O2	9.06	1	PA(TXB2/22:1(13Z))	C45H81O12P	11.21
4	7-Methylguanine	C6H7N5O	6.08	2	PA(PGE1/20:1(11Z))	C43H77O11P	10.69
5	Adenine	C5H5N5	4.01	3	PG(a-21:0/20:4(7E,9E,11Z,13E)-3OH(5S,6R,15S))	C47H85O13P	10.22
6	Oxypurinol	C5H4N4O2	1.72	4	PE(LTE4/20:1(11Z))	C48H85N2O11PS	9.05
7	1-(2-Hydroxyethyloxymethyl)-6-phenyl thiothymine	C14H16N2O4S	1.05	5	PI(18:0/18:2(9Z,12Z))	C45H83O13P	8.81
8	Morph	C10H16N2O4	0.86	6	PA(TXB2/20:0)	C43H79O12P	6.45
9	Deoxyribose	C5H10O4	0.30	7	SM(d16:2(4E,8Z)/20:5 (6E,8Z,11Z,14Z,17Z)-OH(5))	C41H71N2O7P	6.42
10	5-Methyldeoxycytidine	C10H15N3O4	0.20	8	PA(20:4(8Z,11Z,14Z,17Z)/PGF1alpha)	C43H73O11P	6.08
11	Others	-	2.90	9	PG(i-12:0/i-19:0)	C37H73O10P	4.04
Terpene				10	PE(20:5(5Z,8Z,11Z,14Z,17Z)/18:2(9Z,12Z))	C43H72NO8P	3.63
1	Fusarin C	C23H29NO7	57.37	11	Others	-	23.40
2	Ineketone	C20H30O3	14.98	Amines			
3	3-*trans*-*p*-Coumaroylrotundic acid	C39H54O7	13.58	1	Lauryldiethanolamine	C16H35NO2	30.15
4	8-Isobutanoylneosolaniol	C23H32O9	7.65	2	N1,N8-Diacetylspermidine	C11H23N3O2	28.41
5	4a-Methylzymosterol-4-carboxylic acid	C29H46O3	1.61	3	N1,N12-Diacetylspermine	C14H30N4O2	9.58
6	Polyporenic acid C	C31H46O4	0.97	4	N-Acetylcadaverine	C7H16N2O	5.22
7	Methyl lucidenate F	C28H38O6	0.80	5	2-Phenylacetamide	C8H9NO	4.16
8	Auraptene	C19H22O3	0.61	6	N-Cyclohexylformamide	C7H13NO	2.47
9	Eremanthin	C15H18O2	0.52	7	Repaglinide aromatic amine	C22H28N2O4	2.15
10	8-Butanoylneosolaniol	C23H32O9	0.41	8	2-Naphthylamine	C10H9N	2.06
11	Others	-	1.50	9	N-[4-[Acetyl(3-aminopropyl) amino]butyl]-N-(3-aminopropyl) acetamide	C14H30N4O2	1.68
Peptides				10	Fructosamine	C6H13NO5	1.32
1	Pyro-L-glutaminyl-L-glutamine	C10H15N3O5	13.17	11	Others	-	12.80
2	L-beta-aspartyl-L-leucine	C10H18N2O5	11.56				
3	H-Gly-Arg-Gly-Asp-D-Ser-Pro-OH	C22H37N9O10	10.17				
4	Gamma-Glutamylglutamic acid	C10H16N2O7	9.32				
5	Mauritine A	C32H41N5O5	9.03				
6	Isoleucyl-Valine	C11H22N2O3	8.44				
7	Histidyltyrosine	C15H18N4O4	7.17				
8	Glutamylisoleucine	C11H20N2O5	4.87				
9	Bursopoietin	C14H25N7O3	3.50				

Note: ^1^ ET-22-S, secretions of *L. paracasei* ET-22.

**Table 3 molecules-28-01236-t003:** Group setting and culture of biofilm formation.

Groups	Strains	Cultural Conditions
Model	*S. mutans*, PBS	Anaerobic culture at 37 °C for 24 h
ET-22-L	*S. mutans*, ET-22-L	Anaerobic culture at 37 °C for 24 h
ET-22-HK	*S. mutans*, ET-22-HK	Anaerobic culture at 37 °C for 24 h
ET-22-S	*S. mutans*, ET-22-S	Anaerobic culture at 37 °C for 24 h

**Table 4 molecules-28-01236-t004:** The table of primer sequences.

Primers	Forward Sequences (5′–3′)	Reverse Sequences (5′–3′)	Tm (°C)	References
*SpaP*	TGATGTTGCTTCTTCTATGGAG	CAGGTTAGTGTATGTAAGCTGT	53.95	[24]
*gtfB*	CGAACAGCTTCTAATGGTGAAAAGCTT	TTGGCTGCATTGCTATCATCA	55.88	[9]
*RecA*	GCCTATGCTGCTGCTCTTG	TCACCAATATCTCCGTCAATCTC	56.66	[52]
*gbpB*	AGGGCAATGTACTTGGGGTG	TTTGGCCACCTTGAACACCT	56.43	[9]
*gbpC*	TCTGGTTTTTCTGGCGGTGT	GTCAATGCTGATGGAACGCC	56.43	[9]
*gbpD*	TTGACTCAGCAGCCTTTCGT	CTTCTGGTTGATAGGCGGCA	56.43	[9]
*Ffh*	TGGGAATGGGAGACTTGCTTA	GCTCGGAGTTAGGAGGTCAG	57.56	[52]
*relA*	ACAAAAAGGGTATCGTCCGTACAT	AATCACGCTTGGTATTGCTAATTG	55.30	[3]
*comDE*	ACAATTCCTTGAGTTCCATCCAAG	TGGTCTGCTGCCTGTTGC	56.67	[3]
*ldh*	CTTGATACTGCTCGTTTCCGTC	GAGTCACCATGTTCACCCAT	56.54	[3]
*brpA*	GGAGCAGGCCCTCGTTTATT	ATGACCTCACGTTGACGCTT	56.43	[9]
*16S RNA*	CCTACGGGAGGCAGCAGTAG	CAACAGAGCTTTACGATCCGAAA	58.77	[9]

## Data Availability

The data presented in this study are available herein.

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
