# Peer review of "Postbiotics Derived from L. paracasei ET-22 Inhibit the Formation of S. mutans Biofilms and Bioactive Substances: An Analysis"

_molecules, 2023, doi:10.3390/molecules28031236_

Round 1
Reviewer 1 Report
In this article by Zhao et al., the authors investigated the effect of live, heat killed or secretions of Lactobacillus paracasei ET-22 strain on the biofilm forming ability of the oral pathogen Streptococcus mutans. Alongside biofilm formation, they also studied the changes in various gene expressions associated with biofilms. One of the main concerns of the current manuscript is its text which needs thorough revision for grammatical and language correctness. In several occasions, the use of incorrect words or phrases is affecting the scientific understanding of the results and conclusions drawn. Additional concerns that need to be addressed before considering the manuscript further are enlisted:
- Figure 1A – What is plotted in the Y-axis? Is it biomass (i.e., OD600) or crystal violet staining (Abs 595) as indicated in section 4.3? It should be clearly stated in the legend what is plotted in the panels. How did the authors ensure that the planktonic bacteria are not contributing to the result?
- Section 2.2, Line 108 – what do the authors mean by “damaged formation” of biofilms?
- Figure 2A – It will be helpful to have some arrows/traces to indicate the “cauliflower-like” structures within the figure based on what the authors describe in the text.
- Indicate the full form of PI and NO1 when they are used for the first time.
- Section 4.2 - Why was 0.45 um filter used instead of 0.22 um? How did the authors ensure that the data with the "secretions of L. paracasei ET -22" were not affected by the presence of live bacteria? The authors should inoculate some fresh media to ensure the absence of viable bacteria in the secretion.
- Section 4.6 -Indicate the total magnification used for imaging instead of the magnification of only objective lens.
- Table 3 – Describe the specifications of the anaerobic condition used for the bacterial incubations in detail. For example – ratio of the different gases used during incubation.
- Line 69 – what do the authors mean by most used lactobacillus species? Used for what?
- Figure 3C, 5C – Increase the font size/panels to make them easily legible.
- Some examples of the unclear texts are Line 54-55; line 65 -66; line 72, Line 197, 200 – Reword the sentence for grammatical correctness. Line 44 – what is r/min? Line 78 – preliminarily or primarily. “Inhibition effect” should be replaced with inhibitory effect.
Author Response
Dear Reviewer,
Thanks for your comments and suggestions. Your suggestions help to improve the quality of our manuscript. All the changes have been highlighted. Meanwhile, we have finished the revision for grammatical and language correctness with the English pre-editing service provided by MDPI. If you have any other questions or suggestions, please let us know. The details are as follows:
Comment 1: Figure 1A – What is plotted in the Y-axis? Is it biomass (i.e., OD600) or crystal violet staining (Abs 595) as indicated in section 4.3? It should be clearly stated in the legend what is plotted in the panels. How did the authors ensure that the planktonic bacteria are not contributing to the result?
Response:
Thanks for your advice. We totally agree with your opinions. We have changed the Y-axis as “The crystal violet staining reading values, Abs 595”. The reading values can be used to reflect the biofilms formation levels. Our previous Y-axis was not suitable. In the section 4.3, we also have changed the relative expression. More precisely, this indicator was used to evaluate the biofilms formation. In the procedures of crystal violet staining, the culture medium was cleared away. Then the biofilms were washed with PBS. Therefore, the planktonic bacteria were cleared away. This method referred to a previous literature. We had supplemented the washing detail to avoid the mistakes. Please check the changes in Figure 1A, line 484-486, and other else.
Comment 2. Section 2.2, Line 108 – what do the authors mean by “damaged formation” of biofilms?
Response:
Thanks for your suggestions. The “damaged formation” is not appropriate. We have used the “inhibitory effects” to replace the “damaged formation”. Our meaning was to check the inhibitory degree of L. paracasei ET-22 on formation of S. mutans biofilms. Please check this in line 110.
Comment 3. Figure 2A – It will be helpful to have some arrows/traces to indicate the “cauliflower-like” structures within the figure based on what the authors describe in the text.
Response:
Thanks for your suggestions. We totally agree with you. We have supplemented the red and black arrows to respectively indicate the "corolla" and "flower stalk" of the cauliflower-like structure. Please check the Fig 2A and Figure 2 legend in line 130 and 131.
Comment 4. Indicate the full form of PI and NO1 when they are used for the first time.
Response:
Thanks for your suggestions. We have supplemented the full form of PI. Please check this in line 142. However, the N01 does not have the other name. Its full name is N01.
Comment 5. Section 4.2 - Why was 0.45 um filter used instead of 0.22 um? How did the authors ensure that the data with the "secretions of L. paracasei ET -22" were not affected by the presence of live bacteria? The authors should inoculate some fresh media to ensure the absence of viable bacteria in the secretion.
Response:
Thanks for your suggestions. I am sorry that we have written the wrong information about the filters. The aperture is 0.22 um but not 0.45 um. We have changed this. Please see this in line 462.
Comment 6. Section 4.6 -Indicate the total magnification used for imaging instead of the magnification of only objective lens.
Response:
Thanks for your suggestions. This advice is very valuable. We have changed it as total magnification. Please see these changes in line 526 and 527.
Comment 7. Table 3 – Describe the specifications of the anaerobic condition used for the bacterial incubations in detail. For example – ratio of the different gases used during incubation.
Response:
In the original text, we did not describe the method of anaerobic cultivation. We have added relative detail descriptions from line 478 to 480. We did not incubate the bacteria with gas. The anaerobic culture process was finished in the anaerobic culture bags from Hopebio company of China. In this process, the residual oxygen inside the bag was cleared by the AnaeroPack from Mitsubishi Gas Chemical Company, Inc. of Japan.
Comment 8. Line 69 – what do the authors mean by most used lactobacillus species? Used for what?
Response:
Sorry for our unclear description. Lactobacillus is commonly used in probiotic products for digestive tract health. Therefore, we have added the “for digestive system health” in this place. Please check this in line 71.
Comment 9. Figure 3C, 5C – Increase the font size/panels to make them easily legible.
Response:
Thanks for your suggestions. We have increased the font size of Figure 3C, 5C and 5D to make these more recognizable.
Comment 10. Some examples of the unclear texts are Line 54-55; line 65-66; line 72, Line 197, 200 – Reword the sentence for grammatical correctness. Line 44 – what is r/min? Line 78 – preliminarily or primarily. “Inhibition effect” should be replaced with inhibitory effect.
Response:
We have finished the revision for grammatical and language correctness. Please check these in line 56-58, line 67-68, line 74-75, line 205, line 208, and other else. We have changed this “800 r/min” to professional “60 g” in line 461. In original line 78, the “preliminarily” was our real meaning. Because we think our work still need more evidences. We changed this expression as “preliminary investigations” in line 78. We have changed all the “Inhibition effect” to “inhibitory effect” in line 28, 92, 110, 290, 328, 348, and 385.
Best regards,
Ming Zhang and Wei-Lian Hung.
E-mail: zhangming@th.btbu.edu.cn; hongweilian@yili.com.

Reviewer 2 Report
The authors submitted a manuscript entitled " Postbiotics Derived from L. paracasei ET-22 Inhibit Formation of S. mutans Biofilms and Bioactive Substances Analysis " for review. The authors discussed in detail experimentally the effect of the L. paracasei microorganism on the S. mutans microorganism responsible for dental caries. The studied postbiotics showed a high level of inhibiting the development of dental plaque, which may in the future lead to the development of new methods of preventing this dental disease, thus the manuscript is suitable for publication in the journal Molecules, after removing a number of minor errors, listed below:
The authors construct long sentences consisting of a series of abbreviations, which makes it difficult to understand the thought. Use shorter sentences. For example, at lines 90–92, please use short, simple and clear sentences that will allow an MDPI reader to understand the content.
On line 138 please define the abbreviations N01 and PI to improve the usability of the text for MDPI readers.
Line 257, Table 1. At the body of the Terpene is: … 3-O-cis- … , but at International Union of Pure and Applied Chemistry (IUPAC) nomenclature should be … 3-O-cis- … . Comment: It is recommended to use italics for heteroatom symbols and some prefixes for substituent positions and molecular geometry. It is worth following the general trends. Please note the nomenclature throughout the manuscript. For next example, at line 259, Table 2. At the body of the Terpenes, point 3 is: … 3-trans-p- … , should be … 3-trans-p- … , etc. Note, does not apply to commonly accepted proper names for some chemical molecules.
In the body between numbers, a short character is given “ - “, and recently it has become common to use a middle character “ – “. Comment, please correct at lines 145 (twice), 148, 387, 506, 524, 552, and 555. See to lines, such 328, and 346.
At lines 477 and 495 are: … ℃ … . Comment: The cumulative sign of the degree Celsius is given, but in a different style than the whole text, which may hinder editorial work if the manuscript is accepted. Please correct / standardize throughout the manuscript.
At lines 447, 536 (twice), 537, 545, 546, and 550 (twice), please enter the classic degree sign ‘ ° “ in front of the Celsius sign “ C “, see, for example, to line 534. Comment: In the manuscript made available by the editors, the letter “ o “ mimicking degree sign is set too low, is “ oC “, should be “ °C “. Please correct/unify throughout the manuscript.
In the References section, the reference numbers for all references were duplicated. Comment: Only one reference number in each case is enough.
In the references section between page numbers, a short character is given “ - “, and recently it has become common to use a middle character “ – “. Comment, please correct in references such as 1, 2, 4, 6, 10–17, 19, 20, 23, 25, 26, 28–30, 33–35, 38, 43, 44, 46–49, and 53–56. See to lines, such 328, and 346.
At line 648 (ref. 17.), is: … The oral cavity--a key system to understand substratum-dependent bioadhesion on solid surfaces in man. … . Should be … The oral cavity—a key system to understand substratum-dependent bioadhesion on solid surfaces in man. … . Comment: The authors provided an informal hyphen – please use special connecting characters.
Author Response
Dear Reviewer,
Thanks for your comments and suggestions. Your comments and suggestions are professional and have greatly improved the quality of our articles. All the changes have been highlighted. We have finished the revision for grammatical and language correctness with the English pre-editing service provided by MDPI. The details are as follows:
Comment 1: The authors construct long sentences consisting of a series of abbreviations, which makes it difficult to understand the thought. Use shorter sentences. For example, at lines 90–92, please use short, simple and clear sentences that will allow an MDPI reader to understand the content.
Response:
Thanks for your advice. Now, we have finished the revision for grammatical and language correctness with the English pre-editing service provided by MDPI. Hopefully, the revised version will make it easier to follow for you and other readers.
Comment 2: On line 138 please define the abbreviations N01 and PI to improve the usability of the text for DPI readers.
Response:
Thanks for your advice. We have supplemented the full form of PI. Please check this in line 142. However, the N01 does not have the other name. Its full name is N01. We hope this correction can improve the usability of the text.
Comment 3: Line 257, Table 1. At the body of the Terpene is: … 3-O-cis- … , but at International Union of Pure and Applied Chemistry (IUPAC) nomenclature should be … 3-O-cis- … . Comment: It is recommended to use italics for heteroatom symbols and some prefixes for substituent positions and molecular geometry. It is worth following the general trends. Please note the nomenclature throughout the manuscript. For next example, at line 259, Table 2. At the body of the Terpenes, point 3 is: … 3-trans-p- … , should be … 3-trans-p- … , etc. Note, does not apply to commonly accepted proper names for some chemical molecules.
Response:
Thank you for your professional advice. We really ignore these in our writing. Now, we have made modifications according to your comments. Please check these in line 269-Table 1 and line 271-Table 2.
Comment 4: In the body between numbers, a short character is given “ - “, and recently it has become common to use a middle character “ – “. Comment, please correct at lines 145 (twice), 148, 387, 506, 524, 552, and 555. See to lines, such 328, and 346.
Response:
Thanks for your advice. We did ignore these problems. We have corrected these in the new manuscript. Please check all in line 150, 152, 340, 358, 400, 527, 551, 575, 578, and references.
Comment 5: At lines 477 and 495 are: … ℃ … . Comment: The cumulative sign of the degree Celsius is given, but in a different style than the whole text, which may hinder editorial work if the manuscript is accepted. Please correct / standardize throughout the manuscript.
Response:
We are very appreciated of your comments. We have made modifications according to your comments. Please check these changes in line 481, 498, 513, 516, 523, 525, 545-547, 551, 558-560, 569, and 573.
Comment 6: At lines 447, 536 (twice), 537, 545, 546, and 550 (twice), please enter the classic degree sign ‘ ° “ in front of the Celsius sign “ C “, see, for example, to line 534. Comment: In the manuscript made available by the editors, the letter “ o “ mimicking degree sign is set too low, is “ oC “, should be “ °C “. Please correct/unify throughout the manuscript.
Response:
Thank you for your advice. We have changed all the sign to consistent “ °C ”. Please check these in lines 481, 498, 513, 516, 523, 525, 545-547, 551, 558-560, 569, and 573.
Comment 7: In the References section, the reference numbers for all references were duplicated. Comment: Only one reference number in each case is enough.
Response:
Thanks for your advice. We have removed duplicate numbers. Now only one number is left for each reference.
Comment 8: In the references section between page numbers, a short character is given “ - “, and recently it has become common to use a middle character “ – “. Comment, please correct in references such as 1, 2, 4, 6, 10–17, 19, 20, 23, 25, 26, 28–30, 33–35, 38, 43, 44, 46–49, and 53–56. See to lines, such 328, and 346.
Response:
Thanks for your advice. We have changed all the “ - “ between page numbers to the middle character “ – “. Please check these in each reference.
Comment 9: At line 648 (ref. 17.), is: … The oral cavity--a key system to understand substratum-dependent bioadhesion on solid surfaces in man. … . Should be … The oral cavity—a key system to understand substratum-dependent bioadhesion on solid surfaces in man. … . Comment: The authors provided an informal hyphen – please use special connecting characters.
Response:
Thanks for your advice. We have modified these errors. Please check this in line 671.
Best regards,
Ming Zhang and Wei-Lian Hung.
E-mail: zhangming@th.btbu.edu.cn; hongweilian@yili.com.

Reviewer 3 Report
Review comments on “Postbiotics Derived from L. paracasei ET-22 Inhibit Formation of S. mutans Biofilms and Bioactive Substances Analysis”
General comments
It is an interesting research study applying the L. paracasei ET-22 to inhibit the S. mutans Biofilms formation. Authors used multiple methods to detect the growth of S. mutans when adding L. paracasei ET-22 and show the effectiveness of the L. paracasei ET-22. I think the study should be given a mirror revision. Please see the comments below,
1. In fig.1, is that possible to show some indicators in the images, to better understanding the images and changes;
2. The mechanism of the inhibition is not clear, is that possible to give a potential pathway for L. paracasei ET-22 working against S. mutans;
3. I am not sure why amino acids detection is necessary in the study, any relationships with the inhibition? If so, please show more reasons;
4. How you control S. mutans experimental condition, dental condition is very strict.
Author Response
Dear Reviewer,
Thank you for your advice and guidance on the thinking and writing of this paper. With your help, we obviously feel that the quality of the manuscript has been improved. Meanwhile, we have finished the revision for grammatical and language correctness with the English pre-editing service provided by MDPI. All changes and corrections have been highlighted. Please see the details below:
Comment 1: In fig.1, is that possible to show some indicators in the images, to better understanding the images and changes;
Response:
Thanks for your suggestions. According to your comments, we have supplemented some details in fig.1. The images can reflect some changes in the microstructure and compositions of the biofilms. However, more reliable detection methods are needed to accurately quantify some indicators, for example the detections of soluble extracellular polysaccharides (EPS) and insoluble extracellular polysaccharides. In the Fig.1A, we have changed the Y-axis as “The crystal violet staining reading values, Abs 595”. The reading values can be used to reflect the biofilms formation levels. Our previous Y-axis was not suitable. In the section 4.3, we also have changed the relative expression. More precisely, this indicator was used to evaluate the biofilms formation.
Comment 2: The mechanism of the inhibition is not clear, is that possible to give a potential pathway for L. paracasei ET-22 working against S. mutans;
Response:
Thanks for your suggestions. This advice is very valuable. Indeed, we only found the possible functional substances that may mediate inhibition of biofilms, but did not give a possible pathway. Based on previous studies on related issues and our results, we added possible regulatory pathways. We infer that the live L. paracasei ET-22 and its postbiotics may inhibit the ComDE QS pathway to restrain the S. mutans biofilms partly by phenyllactic acids. We have supplemented this. Please check our changes in line 403, 429, and 433.
Comment 3: I am not sure why amino acids detection is necessary in the study, any relationships with the inhibition? If so, please show more reasons;
Response:
Sorry for our unclear description. In our study, amino acids are only one part of the complex postbiotic components, which does not mean that they are involved in the inhibition of biofilms. Actually, sufficient amounts of amino acids may promote the growth of biofilms.
Comment 4: How you control S. mutans experimental condition, dental condition is very strict.
Response:
Thanks for your suggestions. the dental condition is very strict. In order to simulate the S. mutans experimental condition, the artificial saliva medium supplemented with 1% sucrose and hydrox-yapatite discs were added. This experiment was carried out under the anaerobic condition, to mimic oral habits. The artificial saliva medium and sucrose were used to provide a similar nutritional environment to the oral cavity. Meanwhile, the hydrox-yapatite discs were used to simulate the teeth. According to your comments, we have supplemented some experimental details in Section 4.3.
Best regards,
Ming Zhang and Wei-Lian Hung.
E-mail: zhangming@th.btbu.edu.cn; hongweilian@yili.com.

Round 2
Reviewer 1 Report
The manuscript provides insights into the inhibitory effect of live, heat killed or secretions of Lactobacillus paracasei ET-22 strain on Streptococcus mutans biofilm forming ability. The authors have adequately addressing most of the comments and concerns raised during the first review. The revised manuscript has improved both in scientific clarity and its presentation of the experimental results. There are a few typos still present in the text which should be corrected. However, the information presented would be of interest to the readers studying in interspecies bacterial interactions during biofilm formation.